# Evaluation of the HD-GLIO Deep Learning Algorithm for Brain Tumour Segmentation on Postoperative MRI

**DOI:** 10.3390/diagnostics13030363

**Published:** 2023-01-18

**Authors:** Peter Jagd Sørensen, Jonathan Frederik Carlsen, Vibeke Andrée Larsen, Flemming Littrup Andersen, Claes Nøhr Ladefoged, Michael Bachmann Nielsen, Hans Skovgaard Poulsen, Adam Espe Hansen

**Affiliations:** 1Department of Radiology, Centre of Diagnostic Investigation, Copenhagen University Hospital—Rigshospitalet, 2100 Copenhagen, Denmark; 2Department of Clinical Medicine, University of Copenhagen, 2100 Copenhagen, Denmark; 3The DCCC Brain Tumor Center, 2100 Copenhagen, Denmark; 4Department of Clinical Physiology and Nuclear Medicine, Centre of Diagnostic Investigation, Copenhagen University Hospital—Rigshospitalet, 2100 Copenhagen, Denmark; 5Department of Oncology, Centre for Cancer and Organ Diseases, Copenhagen University Hospital—Rigshospitalet, 2100 Copenhagen, Denmark

**Keywords:** brain tumour segmentation, treatment monitoring, routine, postoperative, automatic, deep learning algorithm, external validation, magnetic resonance imaging

## Abstract

In the context of brain tumour response assessment, deep learning-based three-dimensional (3D) tumour segmentation has shown potential to enter the routine radiological workflow. The purpose of the present study was to perform an external evaluation of a state-of-the-art deep learning 3D brain tumour segmentation algorithm (HD-GLIO) on an independent cohort of consecutive, post-operative patients. For 66 consecutive magnetic resonance imaging examinations, we compared delineations of contrast-enhancing (CE) tumour lesions and non-enhancing T2/FLAIR hyperintense abnormality (NE) lesions by the HD-GLIO algorithm and radiologists using Dice similarity coefficients (Dice). Volume agreement was assessed using concordance correlation coefficients (CCCs) and Bland–Altman plots. The algorithm performed very well regarding the segmentation of NE volumes (median Dice = 0.79) and CE tumour volumes larger than 1.0 cm^3^ (median Dice = 0.86). If considering all cases with CE tumour lesions, the performance dropped significantly (median Dice = 0.40). Volume agreement was excellent with CCCs of 0.997 (CE tumour volumes) and 0.922 (NE volumes). The findings have implications for the application of the HD-GLIO algorithm in the routine radiological workflow where small contrast-enhancing tumours will constitute a considerable share of the follow-up cases. Our study underlines that independent validations on clinical datasets are key to asserting the robustness of deep learning algorithms.

## 1. Introduction

### Background

Patients with primary malignant brain tumours have some of the poorest prognoses of all cancer patients even after extensive surgery, radio-, and chemo-therapy [1,2,3]. To monitor treatment effects, magnetic resonance imaging (MRI) is performed regularly. Such minimally invasive response monitoring is of prognostic value and aids therapeutic intervention [4,5,6]. For decades, two-dimensional (2D) measurements of tumour size following international guidelines such as the Response Assessment in Neuro-Oncology (RANO) criteria have been used to evaluate brain tumour treatment response [7,8]. Several studies have, however, indicated that three-dimensional (3D) evaluation of tumour size is more accurate, more reliable [9,10,11,12] and correlates better with prognosis [13]. Tumour size assessment by 2D measurements is still the clinical standard because manual 3D delineation is too time-consuming to be feasible for routine evaluation.

Tremendous progress has been made in order to automate the task of 3D brain tumour delineation by use of deep learning (DL) algorithms as demonstrated by the contributions to the annual Brain Tumor Segmentation (BraTS) Challenges [14]. Yet, brain tumour delineation algorithms are scarce in radiological practice. A likely reason could be the predominant focus on preoperative imaging as well as the lack of external validations on clinical routine datasets to assert generalisability [15]. Preoperative brain tumour segmentation is highly relevant for, e.g., surgical and radiotherapy planning, survival prediction and tumour subtyping as shown in multiple studies [16,17,18]. However, postoperative treatment response monitoring constitutes a major part of the onco-radiological routine. Brain surgery changes the brain morphology by causing, e.g., resection cavities, tissue scarring, haemorrhages and haematomas, and hence postoperative imaging may present challenges for deep learning tumour segmentation algorithms trained exclusively on preoperative datasets [19]. Surprisingly, studies of algorithms intended for postoperative brain tumour segmentation are scarce in the literature. Recently, the deep learning algorithm, HD-GLIO [20,21], enables brain tumour segmentation of patients with glioma having undergone surgery. HD-GLIO provides a tumour segmentation that showed to be a significantly better surrogate endpoint for overall survival than the RANO assessment [20]. HD-GLIO is trained on a heterogenous dataset and tested on a large multi-centre dataset with ground-truth segmentations by expert neuroradiologists, thus addressing the common challenges of deep learning-based medical image analysis [18,22]. For these reasons, HD-GLIO is a state-of-the-art candidate to bring 3D tumour segmentation into routine radiological use. Still, for DL algorithms to reach implementation in radiological practice, algorithms need to be validated on external routine clinical data, preferably by independent parties and in a routine setting, as has been highlighted, e.g., in [23]. 

The aim of this study was to undertake an evaluation of the HD-GLIO brain tumour segmentation algorithm on an external dataset from a consecutive patient cohort that underwent routine MRI for primary brain tumour response evaluation using ground-truth delineations by independent radiologists. Thus, the study addresses the important issue of brain tumour segmentation of patients having undergone surgery and aims to provide evidence for the robustness and generalisability of a state-of-the-art segmentation algorithm.

## 2. Materials and Methods

### 2.1. Patients and Setting

The performance of the HD-GLIO algorithm was tested on a retrospective dataset consisting of consecutive adult patients (aged ≥ 18 years) that underwent brain tumour follow-up MRI during the period 2 July to 16 August 2020. Data were collected on General Electric Sigma (1.5 Tesla) and Premier (3.0 Tesla) MR scanners with standard anatomical sequences (2D axial T1-weighted before [T1-w] and 3D after Gadolinium-contrast infusion [cT1-w], 3D FLAIR and 2D axial T2-weighted sequences). The data are from a single time-point providing one MRI examination per patient. Other than MRI examinations missing these sequences, there were no exclusion criteria.

### 2.2. Study Design

#### 2.2.1. Image Processing

The image processing pipeline is illustrated in Figure 1. The freely available HD-GLIO algorithm was downloaded from [24] and installed on a PC with the Ubuntu 20.04 desktop operating system. Radiologist delineation of contrast-enhancing (CE) tumour lesions and non-enhancing T2-signal hyperintense abnormality (NE) lesions were conducted using the contouring tools in Mirada (Mirada Medical, Oxford, UK) manually and predominantly with the thresholding brush tool. Before import to Mirada, we followed the HD-GLIO guidelines as recommended by the authors [24] for image pre-processing with the following additional steps (see Figure 1):The non-brain-extracted images were registered to the brain-extracted cT1-w sequence.The images were Z-normalised (zero mean and unit variance) based on the voxel values within the brain mask.The voxel values were rescaled to integers and converted from NifTI to DICOM format for compatibility with Mirada.

The delineations were exported from Mirada as files in DICOM-RT Structure Set (RTSS) format and then converted to NifTI binary masks corresponding to the HD-GLIO output format using a custom script.

The image processing was performed on a PC (Intel Core i7-10700K [3.80 GHz] CPU, 64 GB DDR4 SDRAM, NVIDIA Quadro RTX 4000 8 GB graphics card) running the Ubuntu 20.04 desktop operating system. Processing from original DICOM images to HD-GLIO segmentation had a duration of approximately 10 min per subject.

#### 2.2.2. Approach to Ground Truth Segmentation

Reference or ground truth image segmentation of CE tumour lesions and NE lesions followed definitions provided by Kickingereder et al. [20] (pp. 730–731). In particular, NE lesions were “defined as T2-FLAIR hyperintense abnormality excluding the contrast-enhancing and necrotic portion of the tumour, resection cavity, and obvious leukoaraiosis”.

Since the HD-GLIO algorithm is not designed to integrate the clinical context and previous MRI examinations, image findings with contrast enhancement exhibiting potential tumorous characteristics were annotated as CE lesions. For NE lesions, treatment with chemo- and radio-therapy may induce T2-signal hyperintense abnormalities that are not related to tumorous components but are indistinguishable from such. It was decided to not annotate T2-signal hyperintense abnormalities as NE lesions unless contiguous with tumour components or resection cavities in as close accordance with the annotation guidelines from the benchmark BraTS challenges [25,26] as possible (since 2017 the BraTS Challenges have only included pre-operative patient cases, and thus the annotation protocol does not address resection cavities).

Based on separate delineations by a radiology resident (PS) and a board-certified neuroradiologist (JC), consensus delineation was achieved and used for the analysis. A senior neuroradiologist (VL) with 23 years of experience was consulted for a final decision in cases that proved particularly difficult to segment.

#### 2.2.3. Statistical Methods

We assessed the agreement between CE and NE delineations generated by HD-GLIO and corresponding radiologist delineations using Dice similarity coefficients and Bland–Altman plots, and volume agreement was assessed using concordance correlation coefficients (CCCs). The Dice similarity coefficient (Dice) is defined as in [27]:$$\mathrm{Dice}\left(A,R\right)=\frac{\left|A\wedge R\right|}{\left(\left|A\right|+\left|R\right|\right)/2}$$
where $\left|\cdot \right|$ is the segmentation mask, and *A* and *R* refer to the algorithm and radiologist, respectively.

Unless explicitly stated, reported volumes represent the union volumes of the radiologist and HD-GLIO delineations providing a neutral reporting of the delineations in case of disagreement.

#### 2.2.4. Lesion-Wise Inspection

After statistical analysis, we inspected the delineations by HD-GLIO and radiologists in every case and lesion-wise. We studied the disagreements and summarised in which situations they occurred.

## 3. Results

### 3.1. Patients

Sixty-six consecutive MRI examinations of individual patients were retrospectively collected as described in the previous section. Six scans were excluded due to missing sequences; thus, N = 60 scans were included in this study. Patient characteristics are summarised in Table 1. Most patients were with glioblastoma or astrocytoma, had received surgery and radiotherapy (88%), and the majority had received or were receiving 1st line chemotherapy at the time of scan.

### 3.2. Contrast-Enhancing Tumours

Contrast-enhancing tumour lesions were delineated in 31 of the 60 patients by HD-GLIO or radiologists. Size (union volume) ranged from 0.011 cm^3^ to 103 cm^3^. The case presenting with a CE tumour volume of 103 cm^3^ accounted for 44% of the total CE tumour volume across all patients. 18 cases presented with CE tumour volumes of less than 1.00 cm^3^.

In 29 of the 60 patients, the HD-GLIO algorithm and the radiologists agreed that there were no CE tumour lesions. In 19 of the 60 patients, the HD-GLIO algorithm and the radiologists agreed on the presence of CE tumour lesions. In the remaining 12 of the 60 patients, only HD-GLIO (seven patients) or only the radiologists (five patients) identified CE tumour lesions. The CE tumour volumes of those 12 cases were all less than 1.0 cm^3^ (mean 0.22 cm^3^ and range [0.01 cm^3^; 0.93 cm^3^]) and together represent 2.59 cm^3^ or 1.11% of the total CE tumour volume across all patients.

Dice similarity coefficients are shown in Figure 2. The cases of the 12 patients where only HD-GLIO or the radiologists identified CE tumour lesions result in Dice similarity coefficients of zero. For CE tumour volumes, the overall median Dice similarity coefficient was 0.40 (95% confidence interval [0.00; 0.76]) but 0.86 (95% confidence interval [0.61; 0.87]) if excluding the cases with a volume of less than 1.0 cm^3^. For CE tumour volumes less than 1.0 cm^3^, the median Dice similarity coefficient was 0.00 (95% confidence interval [0.00; 0.12]). Figure 3 illustrates an example delineation of both CE and NE lesions.

Tumour volume estimates are shown in Figure 4. A concordance correlation coefficient of 0.997 (95% confidence interval [0.994; 0.998]) was achieved when comparing HD-GLIO’s CE tumour volume estimates with the corresponding radiologist ground truth. The Bland–Altman plot (Figure 5) reveals a low mean of differences and 95% limits of agreement of 0.04 cm^3^ ± 2.81 cm^3^.

### 3.3. Non-Enhancing T2 Hyperintense Abnormality Volumes

NE lesions were delineated in all 60 cases by both HD-GLIO and radiologists, with NE volumes (union volume) ranging between 1.34 cm^3^ and 213 cm^3^ and with a median of 28.1 cm^3^. Dice similarity coefficients are shown in Figure 2. For NE volumes, the median Dice similarity coefficient was 0.79 (95% confidence interval [0.74; 0.82]). A concordance correlation coefficient of 0.922 (95% confidence interval [0.882; 0.948]) was achieved when comparing HD-GLIO’s NE volume estimates with the corresponding radiologist ground truth (Figure 4), with the tendency of HD-GLIO to provide lower estimates, as also reflected by the Bland–Altman plot (Figure 5) revealing a negative mean of differences and 95% limits of agreement of −7.32 ± 21.67. Figure 3 illustrates an example delineation of both CE and NE lesions.

### 3.4. Lesion-Wise Inspection

#### 3.4.1. Contrast-Enhancing Tumour Lesions

We inspected every patient with CE tumour lesion delineation (31 patients) and categorised the differences between delineations by HD-GLIO and radiologists lesion-wise. In 5 of 31 patients, HD-GLIO and radiologists agreed on the presence of all CE tumour lesions, and there were mostly minor disagreements on the spatial extent of the individual lesions.

In 15 patients, one or more CE tumour lesions were not detected by HD-GLIO. All the lesions were small and could be categorised as follows: punctate (occurred in five patients), weakly and diffusely contrast-enhancing (occurred in four patients), with inhomogeneous intensities on the T1-w (occurred in three patients) or as being present in scarred regions (occurred in three patients). Figure 6 depicts an example where HD-GLIO did not delineate a punctate CE tumour lesion. In one patient, the radiologists had misidentified two vessels as CE tumour lesions that HD-GLIO correctly left out.

In nine patients, HD-GLIO delineated one or more contrast-enhancing structures that were not CE tumour lesions. The lesions were mostly small and could be categorised as follows: dural thickening or leptomeningeal enhancement (occurred in three patients), scarred tissue (occurred in three patients), plexus choroideus (occurred in two patients) or as a vessel junction (occurred in one patient). Figure 7 depicts an example where HD-GLIO delineated structures that were not CE tumour lesions. In one patient, HD-GLIO delineated a small CE tumour lesion surrounding a vessel that the radiologists failed to identify. In two patients, it could not be determined with certainty whether a lesion only delineated as a CE tumour lesion by HD-GLIO was in fact a CE tumour lesion.

#### 3.4.2. Non-Enhancing T2 Hyperintense Abnormality Lesions

We likewise inspected the NE lesion delineation in all 60 patients. In general, there was agreement between HD-GLIO and radiologists on the locations of most NE lesions as reflected by the median Dice similarity coefficient of 0.79. However, the following types of repeatedly occurring disagreements were noticed (more than one type of disagreement is possible in each patient): In 19 patients, HD-GLIO missed NE lesioned tissue with relatively lower signal intensities compared to other regions of NE lesioned tissue in the same patient. This was seen as decreased size of the individual NE lesion delineations (occurred in eight patients) but also altogether missed NE lesions (occurred in seven patients). In 18 patients, periventricular regions of NE lesions were delineated only by HD-GLIO (6 patients) or radiologists (12 patients). The impression was that the thicker the periventricular NE lesion, the likelier HD-GLIO was to delineate it. In 6 of 15 patients with NE lesions in subcortical structures, the brainstem or cerebellum, HD-GLIO did not delineate the NE lesions altogether. Figure 3 shows an example of delineations of NE lesions.

## 4. Discussion

In this study, we evaluated the performance of the HD-GLIO algorithm for volumetric segmentation of brain cancer on a consecutive cohort of 66 patients that underwent routine brain tumour follow-up MRI. The algorithm performed very well on NE volumes (with median Dice = 0.79) and CE tumour volumes larger than 1.0 cm^3^ (median Dice = 0.86). However, if considering all cases with CE tumour volume, the performance dropped significantly (median Dice = 0.40). The findings have implications for the application of the algorithm in the routine radiological workflow where small CE tumours will constitute a considerable share of the follow-up cases.

The HD-GLIO algorithm was presented in the study by Kickingereder et al. [20] and tested both on a local and a large multi-centre dataset. The authors reported segmentation with median Dice similarity coefficients of 0.89–0.91 and 0.93 for CE and NE lesions, respectively, across datasets. In the case of 2D instead of 3D T1-w/cT1-w acquisition, the median Dice similarity coefficient for CE tumour volumes diminished to 0.84 [20]. Our results for CE tumour volumes larger than 1.0 cm^3^ are, thus, close to Kickingereder et al. [20]. However, when considering all cases with CE tumour volume, our results differ substantially from these figures.

Only a very limited number of previous studies have evaluated deep learning brain tumour segmentation on external datasets using delineations by independent radiologists as ground truth. Drai et al. [29] evaluated HD-GLIO retrospectively on a population of 42 paediatric patients and found a mean Dice similarity coefficient of 0.67 for CE tumour volumes and a mean Dice similarity coefficient of 0.41 for NE volumes. However, HD-GLIO was not trained on a paediatric population with its potentially different brain and tumour morphology and tumour location, which could explain the modest performance and thus a direct comparison to our results is difficult. The DeepMedic algorithm [30] was trained and evaluated on the longitudinal (pre- and post-operative) BraTS 2015 training set [27] achieving mean Dice similarity coefficients of 0.72 (CE tumour volumes) and 0.90 (whole tumour volumes). Perkuhn et al. [31] independently evaluated the algorithm’s performance on a dataset from consecutive patients with newly diagnosed glioblastomas qualified for resection. The results were comparable to the original study with mean Dice similarity coefficients of 0.78 (CE tumour volumes) and 0.86 (whole tumour volumes). Yogananda et al. [32] trained and validated a deep learning algorithm on the preoperative BraTS 2018 training dataset and achieved mean Dice similarity coefficients of 0.80 (CE tumour volumes) and 0.90 (whole tumour volumes). They subsequently tested the algorithm on different preoperative datasets including an external dataset and achieved slightly reduced mean Dice similarity coefficients of 0.77 (CE tumour volume) and 0.85 (whole tumour volume), respectively. Bouget et al. [33] trained among three algorithms the nnU-Net algorithm [21] using a preoperative multicentre dataset and achieved an average cross-validation mean Dice similarity coefficient of 0.89 for CE tumour volume and enclosed necrotic core. A subsequent evaluation on an external dataset (the preoperative BraTS 2020 training dataset with corresponding independent annotation) yielded a mean Dice similarity coefficient of 0.84 (CE tumour volume and enclosed necrotic core). The group also reported subgrouping by tumour volume from cross-validation during training with mean Dice similarity coefficients of 0.89 (volume ≥ 3 cm^3^) and 0.63 (volume < 3 cm^3^); i.e., as in our study, the algorithm’s performance was greatly reduced on small CE tumour volumes. In summary, previous studies on adults with external evaluation considered the preoperative setting [31,32,33] and not response evaluation, and the performance apparently decreased for small tumour sizes [33]. Our study adds to the current literature by providing an evaluation of the state-of-the-art HD-GLIO segmentation algorithm conducted on a consecutive postoperative patient cohort that underwent routine MRI response evaluation and with independent ground-truth radiologist segmentation.

The patient population of the present study included patients with recent surgery and the majority had no or only small CE tumour volumes. The patient population thus differs from the larger of two test datasets used by Kickingereder et al., the multi-centre EORTC-26101 dataset, where all patients had confirmed first progression of glioblastoma after standard chemo-radiotherapy. Except for Drai et al. [29], the external validation studies mentioned above employed preoperative MRI. We believe our evaluation of HD-GLIO reflects a possible future performance in the radiological routine for treatment response monitoring.

The current study has some limitations. The dataset included MRI examinations of 60 consecutive patients out of which the radiologists identified CE tumour volumes in 24 patients. Yet, the dataset provided a sufficient number of complex cases to demonstrate features that caused disagreement between HD-GLIO and radiologist ground truth delineations. Regarding the ground-truth segmentation, our radiologists delineated CE tumour lesions and NE lesions mainly using a thresholding brush tool. The delineations employed by Kickingereder et al. were carried out using a semi-automatic approach in the segmentation tool itk-SNAP. Here, tumour voxels are classified in a machine-learning step followed by segmentation with active contours [34]. Though both methods are valid, the choice of two different segmentation approaches can be expected to influence the Dice similarity coefficients. Finally, the study only evaluated delineations by HD-GLIO on a single time-point MRI. Therefore, we cannot determine the impact HD-GLIO’s issues with small CE tumour volumes may have on the longitudinal assessment of treatment response. Since small lesions will not contribute much to volume changes, it may be that decreased detection of small CE tumour lesions does not matter much as long as the radiologist is aware of them and report their presence.

In perspective, deep learning-based 3D segmentation of brain tumours has the potential to improve radiological response monitoring and may reduce radiologist workload. The performance of the HD-GLIO algorithm by Kickingereder et al. clearly indicates that automatic volumetry will be the future of brain tumour response monitoring. The results also indicate that it may be beneficial to train the algorithm further, possibly on a dataset acquired with a focus on including patients with small contrast-enhancing lesions. The techniques behind the algorithms transform well between different domains and radiographic modalities [21,35,36]. As a final remark, for training and evaluating segmentation algorithms, consensus guidelines for tumour delineation would be desirable.

## 5. Conclusions

In this study, we evaluated the performance of the HD-GLIO algorithm for volumetric segmentation of brain cancer on a consecutive cohort of patients that underwent routine brain tumour follow-up MRI using independent radiologist ground truth segmentations. The algorithm performed well on the delineation of non-enhancing T2/FLAIR hyperintense abnormality volumes and contrast-enhancing tumour volumes larger than 1.0 cm^3^. However, the performance dropped for the small contrast-enhancing tumour volumes often encountered in treatment response monitoring.

## Figures and Tables

**Figure 1 diagnostics-13-00363-f001:**
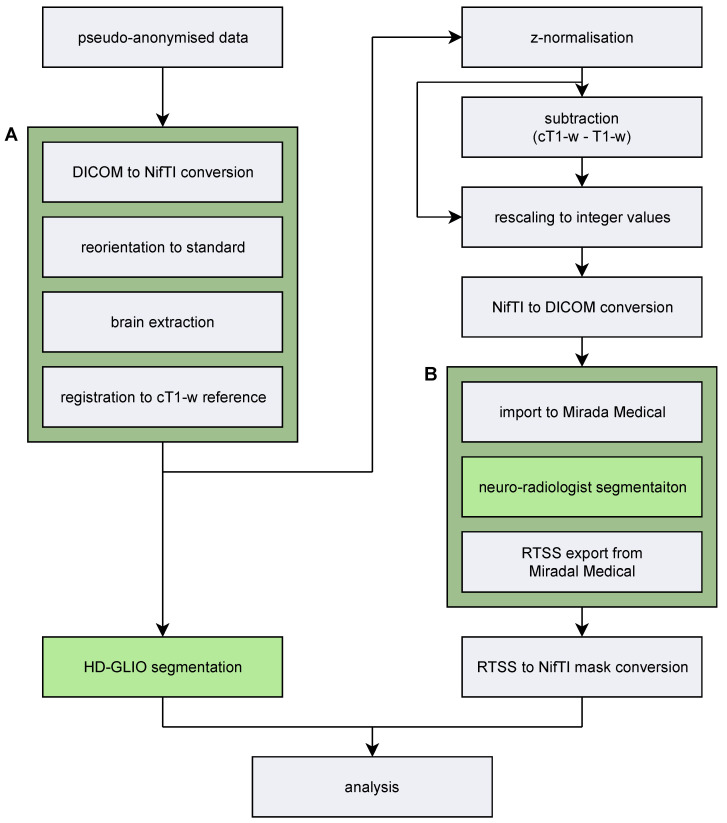
Image processing pipeline. The left column shows the steps prior to HD-GLIO segmentation. (**Box A**) illustrates the general pre-processing steps mandatory for HD-GLIO to work [24]. For optimal resolution, we registered all volumes to the cT1-w reference. The right column shows the steps needed for neuroradiologist-conducted segmentation in the Mirada Medical software. (**Box B**) represents the Mirada software that our neuroradiologists used for segmentation. Segmentations were outputted as contour sequences in RTSS format and converted into NifTI-volume masks similar to HD-GLIOs output files using a custom-made script.

**Figure 2 diagnostics-13-00363-f002:**
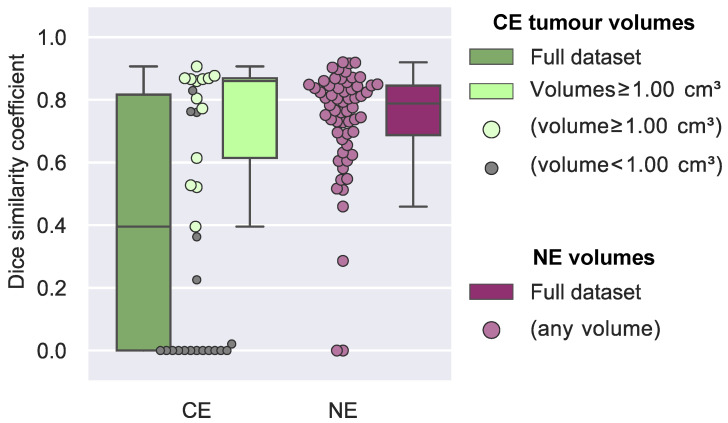
Agreement between HD-GLIO and radiologist delineation of contrast-enhancing (CE) tumour volumes and non-enhancing (NE) T2/FLAIR hyperintense abnormality lesion volumes. The dark green boxplot shows the distribution of Dice similarity coefficients for all CE tumour volumes (median = 0.40), whereas the light green boxplot shows the distribution of Dice similarity coefficients from CE tumour volumes greater than 1.00 cm^3^ (median = 0.86). The purple boxplot shows the distribution of Dice similarity coefficients for NE lesion volumes (median = 0.79). Swarm plots illustrate the Dice similarity coefficient of each case for CE tumour volumes (light green), NE lesion volumes (purple) and volumes less than 1.00 cm^3^ (grey, occurring for CE only).

**Figure 3 diagnostics-13-00363-f003:**
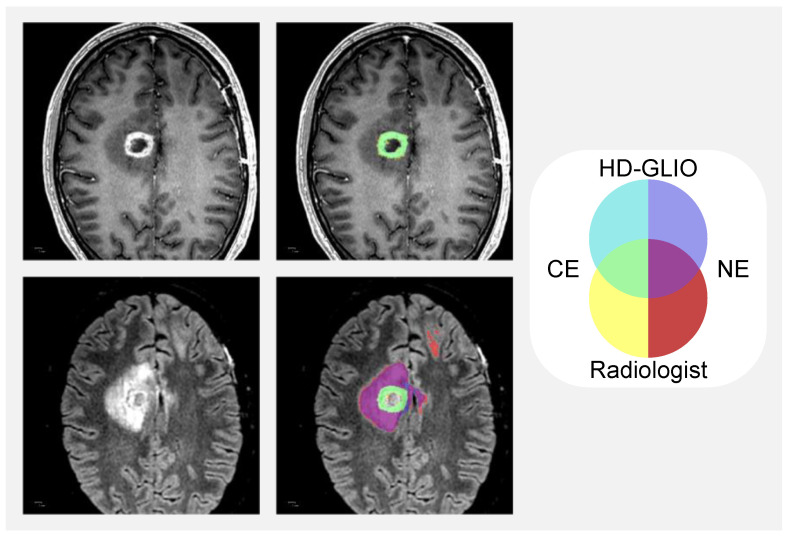
Example of contrast-enhancing (CE) tumour and non-enhancing T2/FLAIR hyperintense abnormality (NE) lesion delineations by HD-GLIO and radiologists for a case with Dice similarity coefficients (Dice) close to the medians of the dataset for tumour volumes >1.00 cm^3^. The left column shows anatomy on T1-weighted MRI after Gadolinium contrast (top) and T2/FLAIR image (bottom). In the right column, the delineations are overlaid semi-transparently. On the top right image, only delineations of CE tumour lesions are overlaid. On the bottom right image, both CE tumour lesions and NE lesions are overlaid according to the shown colour scheme: yellow = Radiologist CE tumour lesion delineation; cyan = HD-GLIO CE tumour lesion delineation; green = overlap of CE tumour lesion delineations; red = radiologist NE lesion delineation; blue = HD-GLIO NE lesion delineation; purple = overlap of NE lesion delineations. CE tumour volume Dice = 0.87; NE volume Dice = 0.81.

**Figure 4 diagnostics-13-00363-f004:**
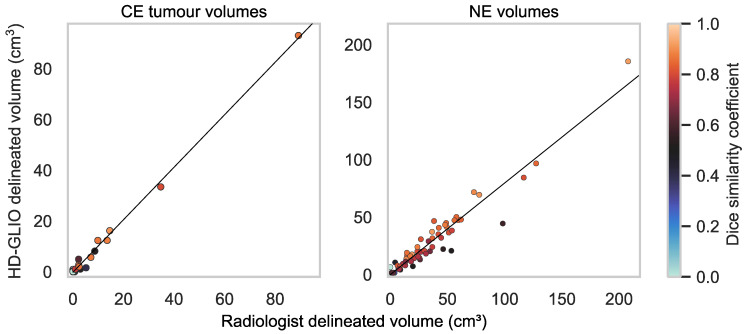
Plot of volume delineation by HD-GLIO against volume delineation by radiologists for contrast-enhancing (CE) tumour and non-enhancing T2/FLAIR hyperintense abnormality (NE) volumes. The scatter markers representing each case are colour-coded by the case’s Dice similarity coefficient. The dashed grey lines represent the line of unity, while the black lines represent the linear trends, with concordance correlation coefficients of 0.997 (CE tumour volumes) and 0.992 (NE volumes). For CE tumour volumes, the correlation is close to the line of unity. For NE volumes, there is a tendency for HD-GLIO to provide lower estimates compared to radiologists.

**Figure 5 diagnostics-13-00363-f005:**
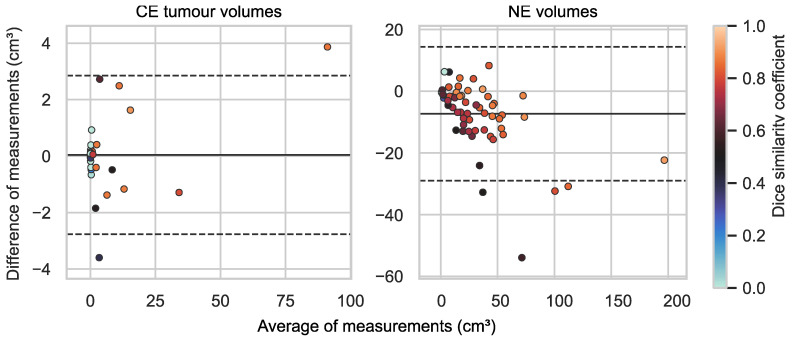
Bland–Altman plots of contrast-enhancing (CE) tumour and non-enhancing T2-signal abnormality (NE) volume estimates by HD-GLIO and radiologists (reference). Each point is colour-coded by its Dice similarity coefficient. For CE tumour volumes the plot shows the mean of differences (solid line) and 95% limits of agreement (dashed lines) of 0.04 cm^3^ ± 2.81 cm^3^. For NE volumes the plot shows a negative mean of differences and 95% limits of agreement of −7.32 ± 21.67 cm^3^.

**Figure 6 diagnostics-13-00363-f006:**
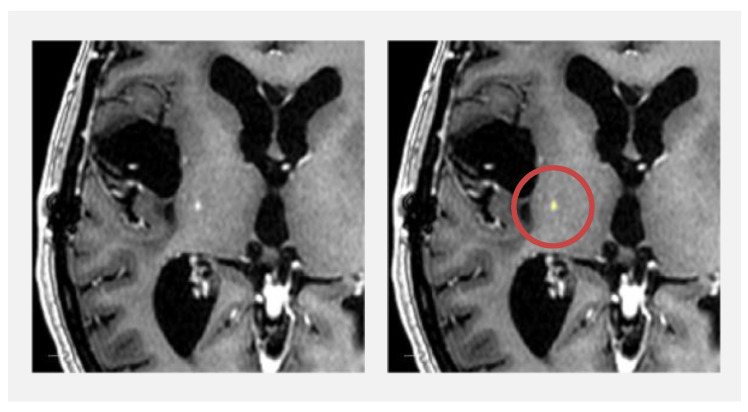
Example of punctate contrast-enhancing (CE) tumour lesion on T1-weighted MRI after Gadolinium contrast that HD-GLIO failed to identify. Right image: overlay with radiologist delineation (yellow). The punctate lesion was the only CE tumour lesion in this patient, and thus CE tumour volume Dice similarity coefficient was zero.

**Figure 7 diagnostics-13-00363-f007:**
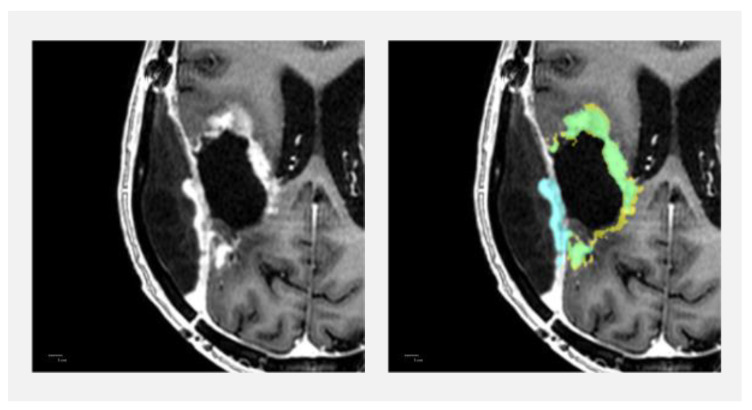
Example case where HD-GLIO identified contrast-enhancing dural thickening as CE tumour while not delineating some of the less enhancing cavity wall. On the right image, delineations of CE tumour lesions are overlaid: yellow = Radiologist CE tumour lesion delineation; cyan = HD-GLIO CE tumour lesion delineation; green = overlap of CE tumour lesion delineations. CE tumour volume Dice similarity coefficient = 0.52.

**Table 1 diagnostics-13-00363-t001:** Patient and scan characteristics.

	Number of Patients (Percentage of Total)
Scans/patients included	Total/1.5 Tesla scans/3.0 Tesla scans 60 (100%)/55 (92%)/5 (8%)
Diagnosis ^1^	Glioblastoma/Astrocytoma/Oligodendroglioma/Other 26 (43%)/17 (28%)/12 (20%)/5 (8%)
Surgery	Prior surgery/no surgery 54 (90%)/6 (10%)
Radiotherapy	Had received radiotherapy/no radiotherapy 53 (88%)/7 (12%)
Oncologic treatment	No oncologic treatment/1st line only/2nd line 8 (13%)/39 (6 5%)/13 (22%)

^1^ Classification according to the 2021 World Health Organization (WHO) Classification of Tumors of the Central Nervous System [28]: Glioblastoma = Glioblastoma, IDH wild type, WHO grade 4. Astrocytoma = Astrocytoma, IDH-mutant, WHO grade 2, 3 or 4. Oligodendroglioma = Oligodendroglioma, IDH-mutant and 1p/19q-codeleted, WHO grade 2 or 3. Other = tumour of unknown origin, non-glioma, glioma with incomplete information from biopsies for 2021 WHO classification and a single case of diffuse hemispheric glioma H3.3 G34-mutated, WHO grade 4.

## Data Availability

Data is not available due to patient privacy and no approval for data sharing has been obtained.

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
