# Peer review of "Evaluation of the HD-GLIO Deep Learning Algorithm for Brain Tumour Segmentation on Postoperative MRI"

_diagnostics, 2023, doi:10.3390/diagnostics13030363_

Round 1
Reviewer 1 Report
The authors have presented a study on the evaluation of the HD-GLIO based DL algorithm. for the automatic segmentation of the brain tumors during routine MRI imaging. The authors have performed an external evaluation on the efficacy of the DL algorithms for brain tumor segmentation in MRI images by considering the dice similarity of CE tumor lesions and NE lesions of the patient and have also evaluated the volumetric analysis by considering the CCC and BA plots.
-
The authors have performed a performance evaluation of the HD-GLIO DL algorithm for the volumetric segmentation of the tumors with consensus from experienced radiologists, and have evaluated the Dice similarity. The model performs well on the tumor lesion > 1 cm2.
-
The authors are invited to further compare their proposed HD-GLIO model with other popular segmentation algorithms for brain tumor segmentation, as well as to evaluate its performance in terms of speed and accuracy.
-
. Additionally, the authors should evaluate their proposed model for smaller tumor lesions < 1 cm2 and compare its results with the manual segmentation.
-
This evaluation would allow the authors to gain further insight into the performance of their proposed HD-GLIO model for smaller lesions and help determine whether or not it is an effective method of segmentation for such small tumors.
-
In some of the lines, there is a typo: "Error!" Reference source not found" (Line, 178, 180, 183, 226–230, 242).
Reviewer 2 Report
1. The paper is weak in terms of establishing the research gap, as introduction section need improvement. Thus, it is strongly advised to include more studies published in well-established journals.
2. Highlight the contributions of this study.
3. Authors must comment how such techniques are helpful when employed on different radiographic imaging for various domains by citing following papers:
- https://doi.org/10.3390/healthcare10050900
- https://doi.org/10.3390/healthcare10071313
4. Must provide the limitations of the study.
Reviewer 3 Report
This paper addresses an important area that interfaces computer science and the medical domain. The paper is well-written and well-structured.
The following improvements are suggested.
1. It would be better to shorten the title.
2. Introduction: The authors have stated the motivation and aim of this study. Please highlight the novel contributions of this study.
3. It is suggested to include a related study section that describes the existing related work. This can be used to identify the existing limitations and challenges and open the path to the proposed study. Such studies can be found in articles such as "Glioma Survival Analysis Empowered with Data Engineering - A Survey", https://doi.org/10.1109/ACCESS.2021.3065965
4. Please improve the scientific contribution of the methodology.
5. Under the discussion section, it would be better to include a table to compare the results with the latest existing studies. (for example, https://doi.org/10.1109/BIBE50027.2020.00014.) This can be used to support the novel contribution of this study.
6. Some references are outdated. For example, consider the 3rd reference from the year 2001. (22 years back). And the title is “Current and future developments ……. of brain tumours”. With the advancements in technology, the authors can find the latest research for current and future trends.
Round 2
Reviewer 2 Report
Thanks. Significantly enhanced. It can be accepted for publication after proofread